# COMET-M: Reasoning about Multiple Events in Complex Sentences

**Sahithya Ravi**[1,2]  **Raymond Ng**[1]  **Vered Shwartz**[1,2]
[1] University of British Columbia
[2] Vector Institute for AI
{sahiravi, rng, vshwartz}@cs.ubc.ca

## Abstract

Understanding the speaker's intended meaning often involves drawing commonsense inferences to reason about what is not stated explicitly. In multi-event sentences, it requires understanding the relationships between events based on contextual knowledge. We propose COMET-M (Multi-Event), an event-centric commonsense model capable of generating commonsense inferences for a target event within a complex sentence. COMET-M builds upon COMET (Bosselut et al., 2019), which excels at generating event-centric inferences for simple sentences, but struggles with the complexity of multi-event sentences prevalent in natural text. To overcome this limitation, we curate a Multi-Event Inference (MEI) dataset of 35K human-written inferences. We train COMET-M on the human-written inferences and also create baselines using automatically labeled examples. Experimental results demonstrate the significant performance improvement of COMET-M over COMET in generating multi-event inferences. Moreover, COMET-M successfully produces distinct inferences for each target event, taking the complete context into consideration. COMET-M holds promise for downstream tasks involving natural text such as coreference resolution, dialogue, and story understanding.

## 1 Introduction

Human understanding of narratives involves building mental representations of the described events (Pettijohn and Radvansky, 2016). We make commonsense inferences about the unstated but plausible causes, effects, and mental states of the participants. This fundamental ability is crucial for NLP systems for human-level performance on tasks such as question answering, summarization, and story understanding. Additionally, reasoning about events can help improve the coherence and consistency of generated text, making it more natural and human-like.

In this work, we focus on deriving event-centric commonsense inferences from sentences with multiple events, prevalent in written and spoken language. Consider an ambiguous news headline such as "Stevie wonder announces he'll be having kidney surgery during London Concert"[1]. Deriving inferences from such sentences is challenging due to the intricate causal and temporal relationships between events. For example, the announcement occurs during the concert, but the surgery happens later. Moreover, it is crucial to stay true to the context when making inferences about specific events.

Existing commonsense knowledge models such as COMET (Bosselut et al., 2019) can generate commonsense inferences along dimensions such as causes, effects and the mental states of the event participants. For example, given the sentence "Stevie Wonder will be having surgery", COMET predicts "he is sick" as a reason. However, since COMET was trained on simple sentences with a single predicate, it falls short on deriving inferences from multi-event sentences, as we show in Fig. 1. COMET's inferences conflate the different events, for example, predicting "he goes to the hospital" as both an event that happened before (the surgery) and after (the announcement). Mapping the inferences back to the events that they refer to is not trivial. In addition, the assumption that the two events happened at the same time leads to incorrect inferences such as "he has to cancel the concert". This hurts COMET's applicability in tasks involving complex sentences.

To address this challenge, we introduce the task of Multi-Event Inference (MEI) which involves generating distinct inferences for each target event within a complex sentence. We collected the Multi-Event-Inference (MEI) dataset consisting of 35k (sentence, target event, dimension, inference) tuples. We use MEI to continue training the latest

---

[1] fox2now.com/news/stevie-wonder-announces-hell-be-having-kidney-surgery-during-london-concert/

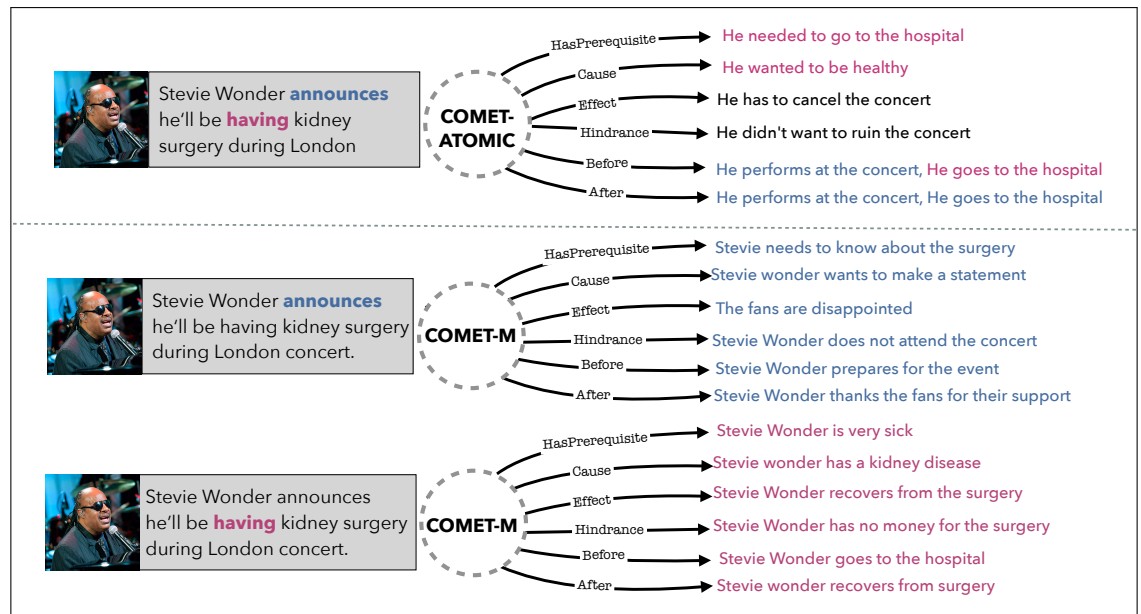

Figure 1: *Top*: COMET's inferences mix up the events announces (inferences in blue) and having [a surgery] (inferences in pink). Inferences in black are incorrect. *Bottom*: COMET-M improves upon COMET by generating inferences for the target events, announces and having. Each set of inferences clearly refers to a single event, e.g. he prepares for the event before the announcement, and rushed to the hospital before the surgery.

version of COMET (Hwang et al., 2021), creating COMET-M, a new multi-event commonsense model. COMET-M can generate distinct inferences for each target event in the sentence, while taking into account the entire context. As shown at the bottom of Fig. 1, when the announces event is the target, our model predicts that he would later thank his fans for their support. Conversely, for the having event, it predicts that he will then recover from the surgery.

COMET-M improved upon COMET in the MEI task in terms of both automatic and human evaluation. Additionally, we found that baselines trained on synthetic multi-event inferences generated from LMs improve over the original COMET in some aspects, but may not be sufficient to reach the same level of performance.

COMET-M has the potential to benefit discourse tasks such as summarization, dialogues and story understanding. Further, MEI can be used to create tasks of varying difficulty for future research.[2]

## 2 Related Work

Reasoning about events is essential for a wide range of NLP tasks such as summarization (Pighin et al., 2014), natural language generation (Chen et al., 2021), dialogue systems (Ghosal et al., 2021, 2022),

and reading comprehension (Rashkin et al., 2018). Earlier approaches represented event types in terms of their participants and subevents, in a structured representation referred to as a "script" (Schank and Abelson, 1975) or "narrative chain" (Chambers and Jurafsky, 2008). For example, a person gets hired at a company, works for it, and then quits, gets fired, or retires. Script knowledge for common events is typically mined from text (Chambers and Jurafsky, 2008; Pichotta and Mooney, 2014; Rudinger et al., 2015; Weber et al., 2020). Knowledge about common events could then be leveraged in order to represent and make predictions about new events.

Later efforts leveraged crowdsourcing to collect large-scale knowledge graphs of event-centric commonsense (Sap et al., 2019; Mostafazadeh et al., 2020). The ATOMIC knowledge graph (Sap et al., 2019) defines events along nine dimensions pertaining to causes, effects, and the mental states of the event participants. It consists of 880k (event, relation, inference) triplets collected through crowdsourcing. The latest version of ATOMIC was extended to 1.1M triplets and additional relations (Hwang et al., 2021). Although ATOMIC is large-scale, it cannot cover the complete range of event-centric commonsense knowledge. Additionally, aligning a given context with the knowledge graph can be challenging, due to lexical variability and because inferences are context-sensitive.

---

[2]Code, models and data are available here.

To that end, Bosselut et al. (2019) developed COMET (COMmonsensE Transformers). COMET is based on pre-trained Language Models (LMs) such as GPT (Radford et al., 2018) or BART (Lewis et al., 2020), which are further fine-tuned on structured knowledge from ATOMIC. As a result, it can dynamically generate event-centric commonsense inferences along ATOMIC's dimensions for new contexts. This hybrid approach is successful because it combines the structure and accuracy of human-written commonsense facts with the vast commonsense knowledge that LMs learn from their massive pre-training corpora (Petroni et al., 2019; Davison et al., 2019). COMET is widely used and has been applied to many downstream tasks such as dialogue and question answering (Kearns et al., 2020; Majumder et al., 2020; Sabour et al., 2021; Kim et al., 2022; Ravi et al., 2023a).

Several subsequent variants of COMET have been released in recent years. COMET-distill (West et al., 2022b) is a version of COMET that is trained on ATOMIC-style events obtained by knowledge distillation from GPT-3 (Brown et al., 2020), while VisualCOMET (Park et al., 2020) generates inferences about the causes and effects of events depicted in an image. Finally, COMET-ATOMIC 2020 (Hwang et al., 2021) is the most recent version of COMET, which was trained on more data and a larger underlying LM.

The variant of COMET most relevant to our work is ParaCOMET (Gabriel et al., 2021b), a paragraph-level COMET model. The key differences between COMET-M and ParaCOMET are as follows. First, COMET-M is trained on various genres for broader applicability, whereas Para-COMET was trained on the ROCStories corpus (Mostafazadeh et al., 2016) in the fiction genre. Second, COMET-M is trained using human-written inferences, unlike ParaCOMET's silver-standard supervision. Our evaluation in Sec 6 demonstrates that silver-standard baselines lag behind the gold-standard baseline in the MEI task on important aspects. Further, COMET-M focuses on generating inferences for multiple events within a single sentence, while ParaCOMET focuses on simple sentences in multi-sentence paragraphs.

In another recent work, Ravi et al. (2023b) improved an existing event coreference system by incorporating GPT-3 generated implicit events that likely occurred before or after each event mention. Differently from Ravi et al. (2023b), COMET-M is

| Relation | Question |
|---|---|
| HasPrerequisite | What are typically the prerequisites for the event? |
| isBefore | What typically happens immediately before the event? |
| isAfter | What typically happens immediately after the event? |
| xReason | What can cause the event? |
| Causes | What could be the effect of the event? |
| HinderedBy | What can hinder the event? |

Table 1: Event-centric relations in COMET-M and the corresponding question templates used for crowdsourcing.

an openly accessible, generic model that can generate inferences along multiple dimensions and is applicable across multiple domains.

## 3 Multi-Event-Inference Dataset

In order to create the Multi-Event-Inference (MEI) dataset, we obtained annotations of target events within complex sentences, through crowdsourcing (Sec 3.2). As the source of the events, we considered datasets from different domains (Sec 3.1).

### 3.1 Events

Within each dataset domain, we extract events from the longest sentences belonging to the 50 most frequent topics.

**News / formal domain.** We use 500 of the gold-labeled event mentions across different topics from the ECB+ dataset (Cybulska and Vossen, 2014), which is a collection of news articles annotated with entity and event coreferences.

**Dialogue.** We sample 200 events from the DREAM dataset (Sun et al., 2019), which includesmultiple-choice questions that were extracted from 6,444 dialogues. For this dataset and the following ones, we use spacy's rule based matching to recognize event mentions by matching contiguous verbs. (Honnibal et al., 2020).

**Narratives.** The SamSum dataset (Gliwa et al., 2019) is composed of 16k chat dialogues with manually annotated summaries. We use the one-line summaries instead of dialogues to add naturally written sentences to our dataset. In addition, we use WikiPlots [3], a collection of plot summaries for more than 42,000 books and movies. We collect annotations for 400 events from Samsum and 200 events from WikiPlots.

---

[3] https://github.com/markriedl/WikiPlots

| | | | |
|---|---|---|---|
| **Instructions:** Read the context sentence and write at least **two** inferences for each question. | | | |
| **Context:** Bryant Dalton was shot and **spent** several weeks at a medical facility. | | | |
| **Question 1:** What are typically the prerequisites of the event spent? | | | |
| **Question 2:** What typically happens immediately before the event spent? | | | |
| **Question 3:** What typically happens immediately after the event spent? | | | |

Figure 2: An example Human Intelligence Task (HIT) on Amazon Mechanical Turk.

**Blogs.** The Webis TLDR Corpus (Völske et al., 2017) corpus consists of 3,848,330 posts from Reddit, with an average length of 270 words. We sample 300 events.

Overall, we collected 1,600 events. To measure the diversity of these events, we calculated the percent of unique bigrams, which was 81%, indicating substantial diversity. In addition, we examined the distribution of the number of events in each sentence across the datasets. We found that 62% of the sentences have two events, 16% contain three events, 13% include four events, 6% feature five events, and 3% include only one event. Finally, to assess the complexity of the events, we sampled 200 multi-event sentences and manually analyzed the temporal relationships between the every pair of subsequent events $E_i$ and $E_{i+1}$ in the sentence. We found the common relationship to be: $E_i$ causes $E_{i+1}$ (35% of the cases), $E_i$ happens before $E_{i+1}$ (24%), $E_i$ happens after $E_{i+1}$ (18%), $E_i$ is the effect of $E_{i+1}$ (16%), and $E_i$ is a prerequisite of $E_{i+1}$ (6%). These results underscore the presence of temporal interdependencies among multiple events in the sentences.

### 3.2 Inferences

We focus on 6 event-centric relations from COMET, presented in Table 1. We collected commonsense inferences for the events we extracted in Sec 3.1 along these relations. The annotation task was conducted on Amazon Mechanical Turk (MTurk). To ensure the annotation quality, we required that workers have completed 5,000 prior tasks with a minimum acceptance rate of 98%. We limited the worker population to native English speakers (US, UK, Canada, Australia, NewZealand) and required workers to pass a qualification test similar to the annotation task. We paid 20 cents for each example or Human Intelligence Task (HIT).

We divided the relations into two sets. In the first set, we asked workers to generate inferences about plausible things that happened immediately before or after the target event, and the prerequisites of the event (rows 1-3 in Table 1). Figure 2 shows an

| | Train 60% | Dev 10 % | Test 30 % |
|---|---|---|---|
| Inferences | 21454 | 3576 | 10726 |
| Target predicates | 919 | 156 | 467 |
| Complex contexts | 557 | 145 | 358 |

Table 2: Statistics of the train, development, and test splits of our Multi-Event Inference (MEI) dataset (Sec 3).

example HIT.[4] In the second set, we asked about the plausible causes, effects, and hindrances of the event (rows 4-6 in Table 1). In both HIT types, we acquired at least two inferences for each relation, and collected inferences from two workers per HIT. On average, we obtained 5 inferences per relation per instance. Overall, we collected 35,418 inferences. We manually reviewed the annotations and removed 2-3% inferences of poor quality, such as incomplete or irrelevant sentences and events annotated as containing offensive or hate speech. We randomly split the data into train (60%), test (30%) and validation (10%) splits, such that the events do not overlap between splits. We show the statistics of events, contexts and inferences in each split in Table 2.

## 4 Model

The goal of Multi-Event Inference (MEI) is to generate commonsense inferences for a target event within a complex sentence. Formally, given a context sentence $C$ that includes $K$ event mentions $E_1, E_2, ..., E_K$, we want to generate a set of commonsense inferences for each event $E_j$ across the 6 inferential relations shown in Table 1. As shown in Figure 1, it is crucial that the inferences generated for each event $E_j$ are *consistent* with the overall context $C$ and *specific* to the event $E_j$.

We build COMET-M based on an existing version of COMET (Hwang et al., 2021) based on BART (Lewis et al., 2020). The motivation is that COMET already captures an abundance of event-centric commonsense knowledge. By continuing to train COMET on the Multi-Event Inference

---

[4]See Appendix A for the full template.

(MEI) dataset Sec 3, we can leverage this knowledge while further adapting the model to generate event-specific commonsense inferences. The training is done with a sequence-to-sequence training objective with the following input and output format:

$$C_i \; R_i \; \texttt{[GEN]} \; T_i$$

where the input is composed of $C$, the context containing event $E_j$ enclosed between `<TGT>` tokens, and $R$, a special token denoting the relation. `[GEN]` is a special delimiter token that indicates the start of the tail of a given relation for a given head entity. Our format is the same as that used in COMET except for the `<TGT>` special token added as a new special token to indicate the event of interest. For consistency with COMET training, we maintain the same relation names. The output $T$ is the target inference in free text.

During inference, we prompt the model in the same format as the training heads, $C_i \; R_i \; \texttt{[GEN]}$

## 5 Experimental Setup

### 5.1 Baselines

We evaluate two types of baselines on the Multi-Event Inference (MEI) test set. The first type consists of off-the-shelf models that are used without any modifications and are not trained on multi-event sentences (Sec 5.1.1). The second type of baselines comprises supervised models that are fine-tuned on multi-event inferences generated automatically, rather than relying on gold standard annotations (Sec 5.1.2).

### 5.1.1 Off-the-Shelf Models

We generate inferences from the following models using the same input format described in Sec 4.

**BART.** We use a pre-trained BART model that was not specifically trained to generate commonsense inferences.

**COMET.** We use the off-the-shelf COMET model from Hwang et al. (2021), which was trained on ATOMIC-2020 but not on multi-event inferences. [5]

### 5.1.2 Supervised Models

We train several baselines with the same setup described in Sec 4, i.e. we initialize the model with

COMET and train it on multi-event inferences. The difference is that rather than training on the gold-standard MEI as in the case of COMET-M, we train the models on silver-standard multi-event inferences generated automatically, as detailed below.

**COMET-M-Split.** This baseline takes complex sentences and automatically splits them into simple sentences in the format COMET is familiar with. Specifically, we take the complex sentences from our training set (Sec 3), and split them into SVO triplets using MinIE (Gashteovski et al., 2017), an Open Information Extraction (OIE) system. As opposed to other OIE systems, MinIE removes overly specific modifiers, yielding simple and short subjects, objects, and predicates. We use the safest mode that eliminates the least amount of unnecessary context. MinIE returns multiple SVO for each complex sentence, with the predicate corresponding to target events. We feed these SVOs into COMET to generate inferences along the various dimensions. The MinIE SVOs along with the events (predicates) and their corresponding COMET inferences serve as silver standard supervision for our task. We fine-tune COMET on this data in the same manner as COMET-M (Sec 4).

**COMET-M-Overlap.** We adopt the same strategy as COMET-M-Split and develop a technique to selectively choose event-specific inferences that are best compatible with the context of the complex sentence. For a given context sentence $C$, and a relation $r$, we first generate the full inferences $I = \{I_1, I_2....I_n\}$ from COMET. Then, to choose the inferences for a target event $E_j$ from this full set $I$, we look at the inferences generated by the MINIE-based split sentence $S_j$ for the same relation $r$: $I^j = \{I_1^j, I_2^j....I_n^j\}$. We pick the inferences from $I^j$ that have a cosine similarity score greater than $0.7$ with any inference in $I$. Since COMET conflates inferences of multiple events, this can be seen as a way to select event-specific inferences that are consistent with the context. We then fine-tune COMET on these selected inferences in the same way as COMET-M. This strategy is similar to the one adopted in ParaCOMET (Gabriel et al., 2021a).[6]

---

[5]Although BART and COMET do not support multiple events, and generate the same inferences for all events within a sentence, we add the `<TGT>` special token to indicate different events.

[6]We don't directly compare with an off-the-shelf ParaCOMET model since it is limited to the fiction domain and uses an older LM.

**COMET-M-NLI.** We follow the same approach as COMET-M-Overlap, but devise a different method to filter out event-specific inferences that are inconsistent with the context. We use a pre-trained NLI model based on RoBERTa (Liu et al., 2019), which was fine-tuned on Multi-NLI (Williams et al., 2018). Given the context sentence $C$, target event $E_j$ expressed in a split sentence $S_j$, relation $r$, and $I^j$, the inferences generated by COMET for the same relation in the simple sentence, we predict the entailment label between $C$ and each inference $i \in I^j$. For most relations, we keep all inferences that are entailed by the context and discard other inferences. For the Hinderdby relation, we keep inferences that contradict the context, since we are interested in scenarios that make the context less plausible. Again, we fine-tune COMET on these filtered inferences in the same way as COMET-M.

**COMET-M-Mimic** Inspired by West et al. (2022a), we generated silver-standard inferences for all MEI training examples by prompting Chat-GPT (gpt-3.5-turbo) with instructions and examples. Similarly to the other baselines, we continue training COMET on this data. The specific instructions and format can be found in Appendix C.

## 5.2 Implementation Details

We train all methods (except the COMET and BART baselines) with the same hyperparameters as the original COMET, listed in Appendix D. During inference, we use beam search with a beam size of 5 and set the maximum decoding length to 50 tokens.

## 6 Results

### 6.1 Automatic Evaluation

Following prior work, we measure the quality of the generated inferences from all models with respect to automatic metrics. We report both BLEU-2 (Papineni et al., 2002) and ROUGE-L (Lin, 2004) scores with respect to the reference inferences. We measure both BLEU and ROUGE as follows. For a given (context, target event, relation) triplet, we remove repetitions from both generated inferences and ground truth inferences. Each generated inference is then scored against all the references. We take the maximum of score with any reference inference for this input. We then average this score across all generated inferences. The intuition is

that, a generated inference is considered correct as long as it overlaps with one of the reference inferences. We also report the average BERTScore (Zhang et al., 2020) which measures semantic similarity between the set of ground truth and generated inferences. As opposed to n-gram overlap based metrics, BERTScore doesn't penalize models for lexical variability.

We present the results of our automatic evaluation in Table 3. In line with the findings presented in COMET (Hwang et al., 2021), a notable disparity in performance is observed between the zero-shot BART and COMET models, emphasizing the value of incorporating explicit learning from commonsense knowledge graphs in addition to pre-training on language.

The COMET-M-Split model demonstrates improved performance compared to the standard COMET model, owing to the fact that it is fine-tuned specifically to generate multi-event commonsense inferences, through the use of automatic labels. COMET-M-Overlap and COMET-M-NLI further exhibit superior performance, as they incorporate methods for filtering out contradictory and irrelevant inferences in relation to the context. Notably, COMET-M-NLI, which explicitly removes inferences that contradict the context, stands out as the best-performing automatic labeling method based on COMET. COMET-M-Mimic performs slightly better than COMET-M-NLI, reassessing previous findings on the effectiveness of using larger model to generate training data for smaller models (West et al., 2022a; Kim et al., 2023). Ultimately, COMET-M surpasses all variants, thanks to its training with human-written multi-event inferences. These results are demonstrated through high scores across all metrics, with a substantial margin from the next best method (e.g. +10 in ROUGE).

### 6.2 Human Evaluation

We assessed the quality of commonsense inferences generated by COMET, COMET-M-NLI (as described in Sec 5.1) and COMET-M (as described in Sec 4) through manual evaluation. Table 4 presents the results, which were obtained by randomly sampling 100 events from the test set. We used the same MTurk qualifications as in Sec 3 and paid 10 cents per HIT.[7] Three workers were presented with a sentence, a target event, and two inferences generated by each model for the rela-

---

[7]See Appendix A for the HIT template.

| Model | Training Data | ROUGE-L | BLEU-2 | BLEU-4 | BERTScore |
|---|---|---|---|---|---|
| BART | - | 11.256 | 4.452 | 1.570 | 50 |
| COMET | Gold standard (simple sentences) | 15.855 | 8.391 | 3.798 | 59.2 |
| COMET-M-Split | Silver standard (multi-event) | 18.071 | 10.074 | 4.681 | 60.5 |
| COMET-M-Overlap | Silver standard (multi-event) | 18.438 | 10.781 | 4.916 | 60.5 |
| COMET-M-NLI | Silver standard (multi-event) | 21.205 | 12.614 | 5.627 | 61.0 |
| COMET-M-Mimic | Silver standard (multi-event) | 23.87 | 14.291 | 6.892 | 61.6 |
| COMET-M | Gold standard (multi-event) | **33.560** | **25.077** | **12.412** | **64.9** |

Table 3: Test performance of off-the-shelf (Sec 5.1.1), silver-standard (Sec 5.1.2) and gold-standard baselines (Sec 4) on Multi-Event-Inference (MEI), with respect to automatic metrics ($\times 100$) computed on the generated inferences and compared to the reference inferences.

| | | %H | %M | %L |
|---|---|---|---|---|
| COMET | Likelihood | 44 | 30 | 26 |
| | Specificity | 26 | 44 | 30 |
| | Relevance | 63 | - | 37 |
| COMET-M-NLI | Likelihood | 45 | 40 | 15 |
| | Specificity | 31 | 53 | 16 |
| | Relevance | 80 | - | 20 |
| COMET-M-Mimic | Likelihood | 43 | 51 | 6 |
| | Specificity | 23 | 66 | 11 |
| | Relevance | 81 | - | 19 |
| COMET-M | Likelihood | 69 | 24 | 7 |
| | Specificity | 62 | 27 | 11 |
| | Relevance | 81 | - | 19 |

Table 4: Human evaluation results for the inferences generated by COMET, COMET-M-NLI, and COMET-M. **H**, **M**, and **L** denote high, moderate, and low.

tions in Table 1. Each inference was rated based on the following aspects:

**Likelihood.** How likely is the event in the given context? Always likely (High-H), sometimes likely (Moderate-M), or never likely (Low-L).[8]

**Specificity.** How specific is the inference to the given target predicate? Highly specific (H), partial overlap with other events (M), or completely about another event (L).

**Relevance.** How relevant is the inference to the entire sentence? Relevant (H) or irrelevant (L).

The results show that COMET-M outperforms COMET, COMET-M-NLI and COMET-M-Mimic in all three metrics. COMET-M-NLI and COMET-M-Mimic show better performance than COMET, but fall short in different aspects.

In terms of likelihood, 69% of COMET-M's inferences were always likely (H), and 24% were

---

[8]We asked raters to mark inferences that simply repeat the target event verbatim as invalid and set low (L) for all metrics for such inferences.

---

sometimes likely (M) with respect to the given relation. COMET, on the other hand, generated low likelihood for the target events in 26% of the cases, which can be attributed to the fact that inferences of multiple events may be conflated. This is also confirmed by COMET's specificity scores being very low.

COMET-M-NLI performs better than COMET on both likelihood and specificity, as it is trained on inferences targeting specific predicates and is less prone to mixing up events. Yet, the inferences are less specific compared to COMET-M, where 62% of the inferences were considered specific to the target event. COMET-M's specificity score leaves room for improvement, with a significant portion of moderately specific inferences that pertain to other events in the context (27%).

COMET-M-Mimic performs on par with the NLI variant on relevance, but fares much worse on specificity, which means that it conflated inferences for different events. Further, we noticed that ChatGPT's inferences that were similar to the context, which is likely a result of its conservative "behaviour" and attempts to not make assumptions that go beyond the provided input.

Finally, all models scored highly in terms of relevance to the given context, with COMET-M achieving a higher relevance score of 81% and COMET-M-NLI closely following with 80%.

## 7 Analysis

### 7.1 Qualitative Analysis on Held-out Corpus

In order to test our model's ability on complex sentences unseen in the collected multi-event corpus, we sampled 25 contexts from a held-out dataset, the DailyDialog (Li et al., 2017). This is a multi-turn dialogue dataset which contains conversations about every day life. We found 3 (out of 5) in-

You see, the bungee participants take a deep breath when they stand at the starting spot and then, like diving, their heads are over heels, and they jump off into the realm of the combination of heaven and earth

|  | stand | take |
|---|---|---|
| COMET | take a breath ✗
they jump off ✓
they fall down ✗
to jump off ✓
they get dizzy ✗ | take a deep breath ✗
they go to heaven ✗
they get dizzy ✓
take a breath ✗
they fall down ✗ |
| COMET-M-Split | they go to the starting spot ✓
they get ready for the race ✗
they get to the starting spot ✓
they get to the finish line ✗
they get in the car ✗ | take a deep breath ✗
to take a deep breath ✗
the bungee falls down ✗
take a deep breath. ✗
the bungee jump ✓ |
| COMET-M-Overlap | to be the best ✗
the bungee jump ✗
to have fun ✓
to be safe ✓
to be happy ✗ | to be safe ✗
the bungee jump ✗
breathe deeply ✗
to have fun ✓
to be happy ✗ |
| COMET-M-NLI | they go to the starting spot ✓
they stand in the starting spot ✓
they are in the starting spot ✓
they get in the bungee ✓ | the bungee participants to take a breath ✗
the bungee participants take a deep breath ✗
the bungee participant to take a breath ✗
the bungee team to take a breath ✗ |
| COMET-M-Mimic | The bungee participants are scared of heights ✗
The bungee participants feel dizzy ✗
They have a fear of heights. ✗
The bungee participants feel exhilarated. ✓ | The bungee participants are scared of heights ✓
The bungee participants feel dizzy. ✓
They have a fear of heights. ✓
The participants are scared ✓ |
| COMET-M | The bungee participants are ready to jump ✓
They want to jump off the bungee ✓
They wanted to jump into the air ✓
They are at the starting spot ✓
They wanted to jump from a height ✓ | The bungee participants are nervous ✓
The bungee participants want to relax ✓
They are nervous ✓
They are jumping high ✓
They feel dizzy ✓ |

Table 5: Inferences generated by different models for the two events **stand** and **take** in a complex sentence taken from the held-out DailyDialog dataset for the relation **xReason** i.e the cause of the event (Li et al., 2017).

ferences to be highly specific to the target event in 90% of the contexts and true to the context in 100% of the contexts. In Table 5, we present the top inferences generated by each of the models for an example sentence from the held-out corpus. As expected, COMET tends to conflate the two events, stand and take, generating similar inferences for both. COMET-Split generates distinct inferences for the two events, but it is apparent from the inference "They get ready for the race" that it doesn't consider the context.

Among the two baselines that filter out inferences, COMET-Overlap also conflates the inferences for the two events, likely because generic inferences such as "to be safe" and "to have fun" are applicable to many events. COMET-NLI performs better by selecting inferences that are consistent with the context, but it tends to repeat the context. This happens because the NLI-based filtering keeps inferences that repeat the context, since they entail it. COMET-M-Mimic performs better in deriving accurate inferences for the take a deep breath event, e.g. deducing that the participants are scared of heights. However, it generated similar inferences for the stand event, demonstrating a lack of specificity, in line with the human evaluation

(Sec 6.2).

Finally, COMET-M demonstrates both specificity and relevance to context by accurately inferring that the reason for the stand event is that the participants are preparing to jump, while the reason for taking a deep breath is that the participants are feeling nervous.

## 7.2 Diversity of multi-event inferences

To further investigate whether inferences generated for different events are semantically distinct, we compare the inferences generated by COMET and COMET-M for different events within the same context.

We focused on the test set and measured the diversity between the sets of inferences generated for each event within the same sentence. To quantify this, we use the ratio of distinct bigrams to all the bigrams in the inferences of all events belonging to the same given context. We obtained a diversity score of 0.65 for COMET-M and 0.47 for the original COMET, confirming that COMET-M can better distinguish the multiple events and generate different outputs for them.

Figure 3 displays a t-SNE projection of the sen-

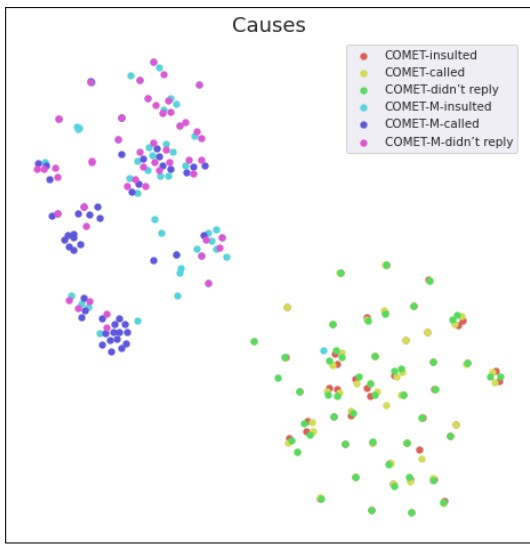

Figure 3: A t-SNE projection of the sentence embeddings of 50 inferences generated by COMET and COMET-M for the Causes relation of the various target events in the sentence "John insulted Mary, so she didn't reply when he called her".

tence embeddings[9] of 50 inferences generated by COMET and COMET-M for the Causes relation of various target events in a sentence with three events: "John *insulted* Mary, so she *didn't reply* when he *called* her". As can be observed, the inferences from COMET-M form more distinct clusters, particularly for the events of *call* and *insulted*. Conversely, COMET generations for the different events overlap significantly.

### 7.3 Usecases for COMET-M

COMET-M can be used to incorporate commonsense knowledge into any task that uses real world text, in particular discourse tasks such as event coreference, story understanding and dialogue. Examples from an unseen dialogue task are explored in Sec 7.1 and the potential of multi-event inferences in solving event coreferences is shown in Ravi et al. (2023b). Hence, we elaborate here on the usecase of story understanding.

**Story Understanding.** Consider the example in Figure 4. The story is composed of two sentences with multiple events. COMET-M can be a valuable tool in unraveling the chain of events that form the story, allowing us to interpret the underlying meaning by reading between the lines. For instance, we can connect the two sentences by understanding

---

⁹paraphrase-MiniLM-L6-v2 model from HuggingFace Sentence Transformers

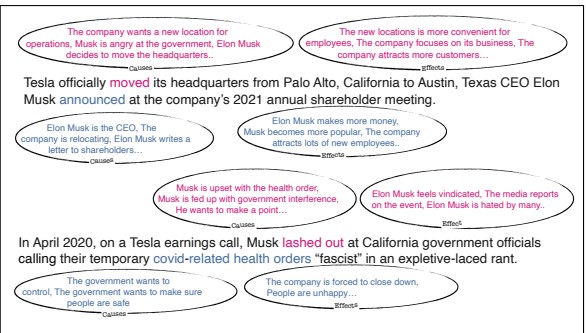

Figure 4: COMET-M can be useful in deriving causes and effects to fill gaps between different lines in a story.

that covid-related health orders caused the company to close down. Similarly, the effects of the moved event, can help in deducing that the company is able to focus more on its business in the new location.

## 8 Conclusions

We focused on the task of generating commonsense inferences for multiple events in complex sentences. To address this task, we proposed a new model, COMET-M, trained COMET-M on human-written inferences that we collected for multiple events within complex sentences. Our experiments, through automatic and human evaluations, demonstrated the effectiveness of COMET-M in generating event-specific and relevant inferences. In the future, we intend to investigate the effectiveness of incorporating COMET-M into discourse tasks such as dialogue, summarization, and story generation.

## Limitations

**Automatic Evaluation Metrics.** Lexical-overlap based automatic metrics were shown to have low correlation with human judgements on various NLG tasks (Novikova et al., 2017). This is especially true for commonsense tasks, which are more open-ended in nature and where various answers may be acceptable. We collected 5-6 reference inferences per relation from human annotators, which cannot cover all plausible inferences for an event.

**Event Mention Detection.** Currently, our approach involves a two-step pipeline where we first identify target predicates, and then generate inferences for those predicates. We plan to simplify this in future, by developing a model that can detect target events and generate inferences for these events.

**Specificity.** Human evaluation (Sec 6.2) revealed that COMET-M generated inferences that were not specific enough to the target event in 27% of cases. We leave it to future work to investigate whether training COMET-M on more data can reduce this.

## Ethical Considerations

**Data.** The datasets used to gather the base events outlined in Sec 3 are publicly accessible. Some of these datasets include information from blogs and forums (e.g. Reddit) which may contain offensive, biased or hateful content. To minimize this, we asked our annotators to label any events they found to be offensive. We excluded such events. Still, we can't fully guarantee the absence of such content from our data.

**Crowdsourcing.** We collected 35k inferences and evaluated 1,200 inferences through crowdsourcing. The participants were compensated with an hourly wage of 16 USD, which is comparable to the minimum wages in the US and Canada. We did not collect any personal information about the participants from MTurk.

**Models.** Our data sources, which include news events and opinions from blogs, may have inherent bias in the commonsense inferences (e.g, based on pronouns such as He/She or country names from news corpora). Our models that are trained on such inferences may produce offensive or biased content for certain events. Pre-trained language models used in our work which are trained on large corpora from the internet may introduce such bias into our models.

## Acknowledgements

This work was funded, in part, by the Vector Institute for AI, Canada CIFAR AI Chairs program, an NSERC discovery grant, and a research gift from AI2.

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

# Appendix

## A  HIT Template

In this section, we provide the templates used for both data collection Figure 5 and human evaluation Figure 6 for the first three relations shown in Table 1.

## B  Input-Output COMET-M

In Table 6 and Table 7 we provide examples from our dataset and inferences generated by COMET-M.

## Task: Pre-requisite, Before and After

> **Context**
> ${context}

### Pre-requisite

**1.** What are typically the prerequisites for the event **${event}**?

(Avoid reusing words from the event)

... ___________________ [write an inference 3-15 words]

___________________ [write another inference 3-15 words]

___________________ [write another inference 3-15 words- optional]

___________________ [write another inference 3-15 words- optional]

### Before

**2.** What typically happens immediately before the event **${event}**?

(Avoid reusing words from the event)

Before ... ___________________ [write an inference 3-15 words]

___________________ [write another inference 3-15 words]

___________________ [write another inference 3-15 words - optional]

___________________ [write another inference 3-15 words- optional]

### After

**3.** What typically happens immediately after the event **${event}**?

(Avoid reusing words from the event)

After ... ___________________ [write an inference 3-15 words]

___________________ [write another inference 3-15 words]

___________________ [write another inference 3-15 words- optional]

___________________ [write another inference 3-15 words - optional]

☐ *Optional Feedback #1:* This event or predicate does not make sense in the given context.
☐ *Optional Feedback #2:* This context/event has hateful/offensive content.
☐ *Optional Feedback #3:* Something about the HIT is unclear/You have additional feedback:

*We plan to post many rounds of these HITs in the near future.*

[ Submit ]

Figure 5: Crowdsourcing template for obtaining before and after inferences.

## C  Prompts Template

In Table 8 we show the prompt used for ChatGPT.

## D  Hyperparameters

The models were trained on a single GPU (NVIDIA GeForce GTX 1080 Ti). We show the hyperparameters in Table 9.nDiversity of inferences We calculate the portion of unique 3-grams out of all 3-grams for the inferences generated per event-relation and report the results in Table 10. We note that silver-standard variants that mix the inferences of other events or deviate the provided context may have inflated diversity scores, as the variability among the 5 inferences may be due to different events. COMET-M exhibits no improvement in diversity score as the 5 inferences for a given event-relation focus on the target event alone. In future work, we aim to explore improving the diversity of generated inferences while remaining specific to the target event.

**Instructions**

Read a context and inferences about what usually happens before and after a target event and the prerequisites for the event. Rate each inference based on how **likely** it is to happen before/after/as prerequisite for the event, how **specific** it is to the event and whether it is **true** to the given context. If you have not tried this task before, please take some time to read the instructions and examples to understand this task better.

**Full Instructions**    (Expand/Collapse)

Thanks for participating in this HIT! In this task, you will be asked to read a context and asked to rate the inferences about a target event.

**For each event, we will provide inferences:**

- 2 inferences about the prerequisite of this event.
- 2 inferences about what typically happens immediately before this event.
- 2 inferences about what typically happens immediately after this event.

**In this task, you will assess these inferences and rate them based on these aspects (questions):**

- How likely is this inference the prerequisite/before/after the event (choose one of: often/sometimes/never/invalid)
- Is the inference true (or relevant) to the context or contradicting (or irrelevant) it? (choose one of: yes/no/invalid)
- Is the inference specific to the target event (choose one of: yes/partially/no/invalid)

**Use the invalid category only for inferences that are not real inferences i.e they just repeat the event.**
The same context can be repeated in different tasks (HITs). This is because the same context can have multiple events, but in each HIT, you will need to focus and rate based on only **one** event.

. The examples below will help you understand how to rate the inferences.

Examples: Target event is in **bold**. (Expand/Collapse)

# Task: Rate event commonsense inferences

**Context**
${context}

## prerequisites

Evaluate these inferences for what NEEDS to happen before, i.e prerequisite of the event **${event}**

${req1}

*How likely is this inference a prerequisite the event?*
always/often    sometimes/ likely    farfetched/never    Invalid(repeats the event)

*Is this inference true to the given context?*
Yes/True to context.    No/negating the context.

*Is this inference specific to the target event ${event}?*
(i.e it does not mix with other events in the same context.)
Yes    Partially specific    Unrelated to the event

${req2}

*How likely is this inference a prerequisite of the event?*
always/often    sometimes/likely    farfetched/never    Invalid(repeats the event)

*Is this inference true to the given context?*
Yes/True to context    No/negating the context.

*Is this inference specific to the target event ${event}?*
(i.e it does not mix with other events in the same context.)
Yes    Partially specific    Unrelated to the event

## Before

Evaluate these inferences for what happens before the event **${event}**

${before1}

*How likely is this inference before the event?*
always/often    sometimes/ likely    farfetched/never    Invalid(repeats the event)

*Is this inference true to the given context?*
Yes/True to context.    No/negating the context.

*Is this inference specific to the target event ${event}?*
(i.e it does not mix with other events in the same context.)
Yes    Partially specific    Unrelated to the event

${before2}

*How likely is this inference before the event?*
always/often    sometimes/likely    farfetched/never    Invalid(repeats the event)

*Is this inference true to the given context?*
Yes/True to context    No/negating the context.

*Is this inference specific to the target event ${event}?*
( i.e it does not mix with other events in the same context.)
Yes    Partially specific    Unrelated to the event

Figure 6: Crowdsourcing template for human evaluation

| Context | Cause(xReason..) | Effects (This Causes...) | HinderedBy |
|---|---|---|---|
| **Human-Written** | | | |
| I set the spray container while I was trimming a bush, [....] the weed killer soaked into the ground near the roses. | The bush was too overgrown. The bush is overgrown and looks unkempt. The bush looks ugly | The person gets thanked for trimming The bush looks more orderly Person is pleased with how the bush looks | The person has no shears for trimming. The bushes were already trimmed The person forgot their hedge clipper |
| I set [...], and the container must have gotten knocked over, and the weed killer soaked into the ground near the roses. | The person knocked the weed killer over Person set weed killer near the roses. The container was not closed.. | The roses will die. The ground gets covered in weed killer. The person can not get rid of weeds | The person closed the lid of weedkiller They paid attention to what they did Person did not buy weed killer |
| **Model-Generated** | | | |
| My lawyer tells me you Ve accepted our alimony proposal and the division of property[..] | They were in a long term relationship They wanted a divorce They have a disagreement in their marriage | The person is happy with the outcome The person is glad they accepted it They say goodbye | My lawyer does not want to help me The alimony proposal was rejected The person does not agree to the alimony |
| My lawyer [..], as well as the custody agreement-I keep the cat and you get the dog | The person wants to keep the cat The person wants to have a pet They had two pets | The cat has a home The person is happy to have the cat The person takes care of the cat | The person does not care about the cat The other person does not want the dog The person does not want the cat |

Table 6: Human-written (top) and model-generated (bottom) examples from Multi-Event COMET for the xReason and Causes (what an event leads to) and HinderedBy relations. Some examples are slightly abbreviated for readability.

| Context | isBefore (Before this..) | isAfter (After this...) | HasPrerequisite (Needed before) |
|---|---|---|---|
| **Human-Written** | | | |
| I set the spray container while I was trimming a bush, [....] the weed killer soaked into the ground near the roses. | He finds the tools He fill his spray container. The person sharpens trimmer | Garden looks nicer Cleans up the bush Bush is trimmed | The person has tools. The weather is good The person has a garden |
| I set [...], and the container must have gotten knocked over, and the weed killer soaked into the ground near the roses. | The container falls The person knocks it The person did not see the container | I snatch the bottle form ground. The roses wilt. The person cleans it up with a rag | They turn around to trim They set the spray container down The container has liquid |
| **Model-Generated** | | | |
| My lawyer tells me you Ve accepted our alimony proposal and the division of property[..] | They discuss the terms of the alimony The person talks to the lawyer The lawyer makes a presentation to the couple | The person is happy with the agreement The person is glad they accepted it They conclude the agreement | They have a lawyer There is legal agreement They need to be married |
| My lawyer [..], as well as the custody agreement-I keep the cat and you get the dog | They both meet They discuss the custody agreement They agreed to divide their property. | The person is happy with the outcome The person is happy to have the cat The person takes care of the cat | The person has a cat The other person wants the dog They have an agreement. |

Table 7: Human-written (top) and model-generated (bottom) examples from Multi-Event COMET for the isBefore, isAfter and HasPrerequisite relations. Some examples are slightly abbreviated for readability.

System Message: You are an expert annotator for commonsense reasoning tasks

User Mesage:

In this task, you will be asked to read a context and asked to write inferences
specific to a target event enclosed between <TGT> tag.
Generate 4 inferences for EACH of the following relations:
isBefore: What typically happens immediately before this event?
isAfter: What typically happens immediately after this event?
HasPrerequisite: What could be the prerequisite of this event?
Causes: What can cause this event to happen?
HinderedBy: What can hinder this event from happening?
Effects: What can be the effect of this event?

Instructions: Inferences are specific to the target event enclosed in <TGT> but
need to stay true to the context. Refer to the examples provided below. Do not mix
inferences of different events in the same context. Provide diverse inferences -
think of many possibilities for a given scenario. You need to generate inferences
for the next example 3. Return the response as a JSON format shown below.

1. The New Bedford man arrested last night on charges of double homicide <TGT>
killed <TGT> his mother and ex-girlfriend, according to police and prosecutors

{"isBefore": ["The man decides to murder them.", "He buys a gun or knife.", "He
follows them into their houses","He keeps watching them"],
"isAfter": ["He hides their bodies.", "He regrets his decision.", "He cleans the
blood from the scene". "He tries to get out of there"],
"HasPrerequisite": ["The man finds his mother and ex-girlfriend", "He needs to
have a weapon.", "He plans the murder". "He has the heart to kill them."],
"Causes": ["The man hates his mother and his girlfriend.", "He is mentally ill",
"He has bad memories with them", "He fought with them many times"],
"Effects": ["The family members of the deceased mourn them.", "The police search
for him", "The man gets caught","It appears on the news"],
"HinderedBy": ["The man could not find them", "He gets nervous" , "He changes his
mind", "They are out of town", "Someone intervenes"]}

2. The New Bedford man <TGT> arrested <TGT> last night on charges of double
homicide killed his mother and ex-girlfriend, according to police and prosecutors

{"Causes": ["Two people are killed", "The police are convinced that he is the
primary suspect", "He is on the CCTV footage", "The police find evidence"],
"HinderedBy": ["The man flees from the country", "The police did not have an
arrest warrant", "They did not find supporting evidence", "The man kills himself
after"],
"Effects": ["The man goes to jail", "The family appreciates the police for their
timely arrest.", "The media applaud the police", "He goes to prison"],
"HasPrerequisite": ["The police need an arrest warrant.", "He must not escape.",
"The police chase after the suspect.", "The police receive a complaint"],
"isAfter": ["The man is taken to the station","The man calls his lawyer", "The
police hold a press conference","The police talk to prosecutors"],
"isBefore": ["The police chase after the suspect", "The police receive a
complaint", "The police find the bodies", "The police take out their handcuuffs"]}

Table 8: Prompt used for ChatGPT.

| Parameter | Value |
|---|---|
| Epochs | 2 |
| Batch Size | 8 |
| Learning Rate | 1e-5 |
| Optimizer | AdamW |

Table 9: Hyperparameters used by all supervised model versions.

| Model | Distinct-3-grams |
|---|---|
| COMET | 62 |
| COMET-M-NLI | 63 |
| COMET-M-Mimic | 64 |
| COMET-M | 63 |

Table 10: Distinct 3-grams for a given event and relation among the 5 inferences generated by beam search