# OpenReview forum: "COMET-M: Reasoning about Multiple Events in Complex Sentences"
_EMNLP/2023/Conference — EMNLP 2023 Findings_

### Official Review · Reviewer_QbAU · 2023-07-24

**Soundness:** 3

**Excitement:**

4: Strong: This paper deepens the understanding of some phenomenon or lowers the barriers to an existing research direction.

**Paper Topic And Main Contributions:**

The main contribution of this paper is the authors have curated a Multi-Event Inference (MEI) dataset of 35K human-written inferences. The MEI dataset is designed to overcome the challenges of complex reasoning in multi-event sentences

**Reasons To Accept:**

The authors have demonstrated the effectiveness of their dataset by evaluating several baseline models and showing that their proposed models outperform the baselines. Overall, the paper is well-written and presents a clear and comprehensive description of the dataset.

**Reasons To Reject:**

While the paper presents a valuable contribution by curating an annotated multi-event sentence dataset, I would like to see more experimental analysis of the dataset. Specifically, it would be interesting to see how human annotation considers the relationship between multi-events, as this would provide insights into the complexity of the dataset and demonstrates why COMET fails. Additionally, I would like to see a more detailed analysis of why fine-tuning BART on the dataset performs better than ChatGPT-prompted answers, especially given that ChatGPT has been shown to be proficient in commonsense reasoning. I would like to raise my scores if my consideration could be solved.

**Reproducibility:**

4: Could mostly reproduce the results, but there may be some variation because of sample variance or minor variations in their interpretation of the protocol or method.

**Reviewer Confidence:**

3: Pretty sure, but there's a chance I missed something. Although I have a good feel for this area in general, I did not carefully check the paper's details, e.g., the math, experimental design, or novelty.

---

> ### Author Rebuttal · Authors · 2023-08-29
>
> We thank the reviewer for the useful feedback and questions.
>
> **Why fine-tuning BART on the dataset performs better than ChatGPT-prompted answers, especially given that ChatGPT has been shown to be proficient in commonsense reasoning.I would like to raise my scores if my consideration could be solved.**
>
> That is a good question. Regarding ChatGPT’s performance on reasoning tasks, it is important to note that we did not finetune ChatGPT, but used the responses from a 6-shot ChatGPT [shown 6 in-context examples]  to train COMET.  This means that we are not evaluating ChatGPT's ability to perform MEI, but we are the ability of a smaller model [here BART] that uses ChatGPT as the annotator instead of a human. The reason why this automatically labeled baseline fails compared to human annotations is two-fold. First, though ChatGPT has been shown to be proficient in commonsense reasoning, it lacks training on the task structure and specific relations of MEI/COMET.  A supervised version of ChatGPT is expected to perform at least similarly, or even better, than the BART-based versions. Second, simply imitating a large model’s responses (ChatGPT) may result in small models (BART) mimicking the style of the larger model, but not its reasoning abilities [1]. So it may be a failure of BART to effectively learn from ChatGPT responses.
>
> **Why COMET fails on MEI**
>
> With respect to COMET’s poorer performance on MEI- COMET is primarily trained on single-event sentences and mixes up the inferences for different events when presented with a multi-event sentence as shown in Figure 1.  We further show the intricacy of multi-event sentences in our results in Table 2 and analysis in Table 4. We show that COMET does not perform well on the MEI task, even when we decompose the sentence and feed it to COMET (COMET-M-Split and COMET-M-NLI). We believe that this empirically shows why COMET is not designed to work with multi-event sentences.
>
> [1] Gudibande, Arnav et al. “The False Promise of Imitating Proprietary LLMs.” ArXiv abs/2305.15717 (2023): n. pag.

---

### Official Review · Reviewer_VGTR · 2023-08-03

**Soundness:** 3

**Excitement:**

4: Strong: This paper deepens the understanding of some phenomenon or lowers the barriers to an existing research direction.

**Missing References:**

1.	Muhao Chen, Hongming Zhang, Haoyu Wang, and Dan Roth. 2020. “what are you trying to do?” semantic typing of event processes. In Proceedings of the 24th Conference on Computational Natural Language Learning (CoNLL 2020). Association for Computational Linguistics.
2.	Li Zhang, Qing Lyu, and Chris Callison-Burch. 2020c. Reasoning about goals, steps, and temporal ordering with WikiHow. In Proceedings of the 2020 Conference on Empirical Methods in Natural Language Processing (EMNLP), pages 4630–4639, Online. Association for Computational Linguistics.

**Paper Topic And Main Contributions:**

This paper proposes COMET-M to generate commonsense inferences for multiple target events within a complex sentence. To overcome the limitation of lacking data about multi-event sentence annotations, the authors curate a Multi-Event Inference (MEI) dataset of 35K human-written inferences. They train COMET-M on MEI and create baselines using automatically labeled examples by prompting Chat- 388 GPT (gpt-3.5-turbo) with instructions and examples. The experiments, through automatic and human evaluations, demonstrated the effectiveness of COMET-M in generating event-specific and relevant inferences.

**Questions For The Authors:**

How to extract the events in a complex sentence? Is the lexical information of events are taken into consideration and how? Do the relations of events affect the event-centric reasoning even the same context is given? Will COMET-M benefits from both silver standard and gold standard annotations?

**Reasons To Accept:**

1.	The reasoning of multi-event complex sentences is an important problem in language understanding and generation.
2.	This paper creates a new dataset with human-written inferences corresponding to multiple events in long sentences collected from news, dialogue, narratives, and blogs.
3.	The proposed model COMET-M show the effectiveness of event-centric annotations on commonsense reasoning of complex sentences.

**Reasons To Reject:**

1.	Description of the motivation is not very clear. It would be better if the authors give a formal formulation and concrete example about what the events are. It is not very clear about the impacts of lexical relation between events and the contextual information inner sentences on the annotations of inferences.
2.	The details about the dataset are missing to measure the coverage, distinction, balance, and key statistical characteristics of it.
3.	The baseline models need to be elaborated more. Besides the off-the-shell models, BART and COMET, all other supervised models are initialized the with COMET and then trained on multi-event inferences. It seems like finetuning COMET to adapt to the downstream tasks. It might make the comparison between COMET and COMET-M unfair because the data distributions of them are different. A randomly initialized model trained on the proposed dataset MEI needs to be considered as a baseline to confirm the effectiveness of the annotations about multi-event inferences.

**Reproducibility:**

2: Would be hard pressed to reproduce the results. The contribution depends on data that are simply not available outside the author's institution or consortium; not enough details are provided.

**Reviewer Confidence:**

4: Quite sure. I tried to check the important points carefully. It's unlikely, though conceivable, that I missed something that should affect my ratings.

**Typos Grammar Style And Presentation Improvements:**

In section 4, the format of input and output needs to be clarified.
Figure 4 is difficult to read, is there any statistics of the repetition of words or n-grams counts to show the diversity of multi-event inferences.

---

> ### Author Rebuttal · Authors · 2023-08-29
>
> Thank you for acknowledging the importance of reasoning in multi-event complex sentences and for the helpful feedback .
>
> **How to extract the events in a complex sentence?**
>
> **Formal formulation and concrete example about what the events are.**
>
> Thanks for pointing this out. In this paper, events refer to multiple target predicates in a sentence. It is the equivalent of verbs obtained by POS tags or semantic role labeling. In this sentence: “If you liked the music we were playing last night, you will absolutely love what we're playing tomorrow!”, the four events would be: liked, were playing, love, ‘re playing.
> The target events/predicates are extracted by finding the verbs in the sentence. This can be achieved by semantic role labeling, information extraction or part-of-speech detection.  We use a simple rule-based matching algorithm based on parts-of-speech tags [1] to find specific patterns based on the parts of speech (e.g auxiliary verbs followed by verbs such as “is watching”, verbs followed by prepositions such as “bring about”). We will include the rules used to extract events in the paper.
>
> **Is the lexical information of events are taken into consideration and how?**
>
> We explicitly instruct humans to consider the entire context of the complex sentences to write inferences. When annotators are asked to label event inferences, we believe that they consider the lexical information, because they interpret and understand the events based on the words and language used to describe them.
>
> **Given the same context, the relations of events will affect the inferences.**
>
>  Could you please elaborate on this question?
>
>
> **Will COMET-M benefits from both silver standard and gold standard annotations?**
>
> That is a great question. Our experiments on combining the NLI-based silver standard annotations with gold-standard resulted in worse performance [e.g ROUGE=32.65]. Likewise, combining ChatGPT inferences results in no performance gain w.r.t automatic metrics [e.g ROUGE = 33.53].  This is also reflected in the human evaluation, where the silver standard based models perform poorly, especially on specificity. We aim to explore how to filter the obtained silver standard annotations or perform more complex data distillation from larger models in future work.
>
> **A randomly initialized model trained on the proposed dataset MEI needs to be considered as a baseline to confirm the effectiveness of the annotations about multi-event inferences.**
>
>  We trained a randomly initialized BART model on the proposed dataset MEI and obtained a ROUGE score of 26.3 points.  This model performs better than COMET on the task of MEI, but worse than the model initialized with COMET and further trained on MEI. Given that COMET is trained on millions of commonsense triplets, this result is expected, and this was our motivation to perform transfer learning from COMET instead of BART.
>
>  **Figure 4 is difficult to read, is there any statistics of the repetition of words or n-grams counts to show the diversity of multi-event inferences.**
>
> We will include the missing references and a better version of Figure 4 in the paper. Following your suggestion, we quantitatively measured the diversity of multi-event inferences in the test set, using the ratio of (distinct bigrams/all possible bigrams) of inferences across events belonging to the same given context.  We obtained a diversity score of 0.65 for COMET-M and 0.47 for the original COMET. This indicates that there is higher diversity in the inferences across events of the same context, for COMET-M compared to COMET. This is in line with what we try to demonstrate in Figure 4. We will include this in the paper.
>
> **The details about the dataset are missing**
>
> Thanks for pointing out that we need to include more details and statistics about the MEI dataset. The target predicates have a diversity score of 0.81, obtained by counting the number of unique bigrams across all bigrams. While a majority i.e 62% of our sentences have two events, MEI includes 16% complex sentences with three events, 13% sentences with four events, 6% sentences with five events and 3% with a single event.  We sample our dataset from different domains to ensure topic diversity.  Given the extra page for the camera-ready version, we will include the statistics as well as an analysis of the topics covered in the paper.
>
> [1] https://spacy.io/api/match

---

### Official Review · Reviewer_2nKM · 2023-08-07

**Typos Grammar Style And Presentation Improvements:** 1. Line 47, "Stevie Wonder"
2. Line 2…
**Soundness:** 2

**Excitement:**

4: Strong: This paper deepens the understanding of some phenomenon or lowers the barriers to an existing research direction.

**Paper Topic And Main Contributions:**

The paper introduces a new human-annotated resource, the Multi-Event Inference dataset. This dataset mainly extends the setup of COMET with more specific event control on more complex sentences. Four domains are covered in the dataset, totaling around 1.5K events. Experiments mainly show the quality of the data is better than decompose complex sentences or using GPT3.5-turbo as data annotator.

**Questions For The Authors:**

A. Have you experimented with not continuing training on COMET?
B. Any results that trained with the larger model? Such as fine-tuning GPT3.5/4, or LLaMA?

**Reasons To Accept:**

1. An exciting artifact that improves the previous model, COMET's drawback --- does not have detailed enough control on the event.
2. I can foresee the potential contribution of the paper to enhance downstream tasks. As for real-case NLP, reasoning on complex sentences is a much more important issue we need to approach.

**Reasons To Reject:**

1. The analysis of the dataset is weak. We would suggest at least providing these in the paper:
a. How diverse are the events in the dataset? (In line 195, you specify the top 50 topics; what are they? Why only cover 50?) How many types are there? How complex are the sentences in the dataset? Are there mainly only two events in the dataset?
b. What are the failure cases in your COMET-M model based on your human evaluation's feedback?
c. How much different will your model react for the same target event but paired with different co-occurred events? For example, comparing "Stevie Wonder announces he'll be having kidney surgery during London concert" to "Stevie Wonder announces he'll get married during London concert." Understanding this can help the reader understands more about how the event's interaction influences the predictions.

2. The dataset size is small (in terms of event counts), and the relation coverage is limited compared to COMET.

**Reproducibility:**

2: Would be hard pressed to reproduce the results. The contribution depends on data that are simply not available outside the author's institution or consortium; not enough details are provided.

**Reviewer Confidence:**

3: Pretty sure, but there's a chance I missed something. Although I have a good feel for this area in general, I did not carefully check the paper's details, e.g., the math, experimental design, or novelty.

---

> ### Author Rebuttal · Authors · 2023-08-29
>
> We thank the reviewer for the insightful feedback, and for identifying the potential contribution of the paper to enhance downstream tasks.
>
> **A. Have you experimented with not continuing training on COMET?**
>
> Thanks for suggesting this interesting ablation. We retrained COMET-M by initializing with BART instead of COMET and obtained a ROUGE of 26.2.  While this baseline still improves upon COMET by ~10 ROUGE points, it lags significantly behind the COMET-M baseline initialized with COMET (ROUGE = 33.5). This is expected, since, COMET is trained on more than one million commonsense triplets, and is exposed to the task format as well as the event-relations. Hence, transfer learning from COMET instead of from BART is beneficial for the MEI task. We will include this baseline as an ablation in our paper.
>
> **B. Any results that trained with the larger model?**
> Since COMET is still widely used in commonsense reasoning research [1, 2], we aimed to improve upon it.  For assessing how to enhance COMET on Multi-Event-Inference, we compared against COMET trained on the responses generated from GPT 3.5 instead of directly fine-tuning GPT 3.5. Given the potential of larger LLMS such as GPT 3.5 (or Llama), we expect their fine-tuned versions to perform at least similarly, if not better, than COMET-based baselines. With the fine-tuning support released for GPT 3.5 recently, we will include a baseline fine-tuned on gpt3.5 (or llama2) in the final paper. We also note that the released MEI dataset can be used to train/evaluate any of the larger models as future work.
>
> **#1a. How diverse are the events in the dataset? (In line 195, you specify the top 50 topics; what are they? Why only cover 50?) How many types are there? How complex are the sentences in the dataset? Are there mainly only two events in the dataset?**
>
> Thanks for highlighting the need to add additional details about MEI dataset.  To answer your question on event diversity, we measured the diversity of 1.5K  target predicates by counting the number of unique bigrams across all bigrams. We found the diversity score at 0.81, indicating substantial diversity. Regarding event distribution across sentences, out of 557 unique complex sentences, 62% have two events, 16% contain three events, 13% include four events, 6% feature five events, and 3% include only one event.  We will include this in the paper.   Regarding L195, the topics are pre-defined in the original datasets. For instance, in ECB+ dataset, the topics refer to different seminal events, while in the reddit dataset, they refer to different subreddits. We decided to pick events from the top 50 most frequent topics to represent each dataset, and make the annotation process more manageable and efficient.
>
> **#1b. What are the failure cases in your COMET-M model based on your human evaluation's feedback?**
>
> Humans found 11% of the inferences from COMET-M to be non-specific to the target event. To demonstrate a failure, let us consider the same example: Stevie Wonder “announces” he'll be having kidney surgery during a London concert. Though the majority of the inferences are specific, a failure occurs when COMET-M predicts “Stevie Wonder needs to be very sick” as one of the prerequisites for both the ‘announces’ and ‘kidney surgery’ events.
>
> **#1c. How much different will your model react for the same target event but paired with different co-occurred events? For example, comparing "Stevie Wonder announces he'll be having kidney surgery during London concert" to "Stevie Wonder announces he'll get married during London concert."**
>
> We generated the inferences of "Stevie Wonder announces he'll get married during London concert." to see how different they are from “Stevie Wonder announces he'll be having kidney surgery during London concert" [Figure 2]. COMET-M predicts that, “Stevie Wonder prepares a speech” before the ‘announces’ event, and that, ‘Stevie Wonder gets lots of congratulations’ after.  For the ‘getting married’ event, COMET-M predicts that “Stevie Wonder buys a wedding ring” before, and that “He is very happy” after.  This indicates that the event interactions influence the predictions accurately.
>
> **#2. Dataset size**
>
> With respect to dataset size, we believe that our dataset size of ~1.5K events and ~45K commonsense triplets is reasonable, especially considering that we are transfer learning from a strong commonsense model i.e., COMET which is trained on 1.1M triplets. We focus on six important event-based relations pertaining to causes, effects, hindrances, and prerequisites of events, and leave it to future work to investigate object-oriented relations.
>
> [1] Cui, Wanyun and Xingran Chen. “Free Lunch for Efficient Textual Commonsense Integration in Language Models.” Annual Meeting of the Association for Computational Linguistics (2023).
>
> [2] Zandie, Rohola et al. “COGEN: Abductive Commonsense Language Generation.” Annual Meeting of the Association for Computational Linguistics (2023).

---

### Meta-Review · Area_Chair_1iMk · 2023-09-20

**Recommendation:** 4

**Metareview:**

The paper presents a new dataset for multi-event inference on complex long sentences. This is an addition to the line of common-sense inference resources such as COMET and ATOMIC. The key contributions include the resource itself, a demonstration of the utility of the resource on the proposed task.

The main problem of wanting to address reasoning over complex sentences and the gap with respect to an existing resource like COMET is well motivated. The solution methodology of obtaining a manually curated resource is reasonable. The resulting resource is likely valuable. The paper is well written overall with the caveat of the lack of some details and motivations for the design choices, as identified by the reviewers. The originality of the work lies in stating and tackling a problem that hasn't been addressed yet. The methodology is sound overall and the resulting resource is of high quality. The scope and scale of the resource however are not as big as that of prior resources on similar problems. This is hard task to create resources for and the authors make a reasonable empirical case for the utility of the resource.

---

### Decision · Program_Chairs · 2023-10-07

**Decision:**

Accept-Findings

**Comment:**

The paper presents a new dataset for multi-event inference on complex long sentences. This is an addition to the line of common-sense inference resources such as COMET and ATOMIC. The key contributions include the resource itself, a demonstration of the utility of the resource on the proposed task.

The main problem of wanting to address reasoning over complex sentences and the gap with respect to an existing resource like COMET is well motivated. The solution methodology of obtaining a manually curated resource is reasonable. The resulting resource is likely valuable. The paper is well written overall with the caveat of the lack of some details and motivations for the design choices, as identified by the reviewers. The originality of the work lies in stating and tackling a problem that hasn't been addressed yet. The methodology is sound overall and the resulting resource is of high quality. The scope and scale of the resource however are not as big as that of prior resources on similar problems. This is hard task to create resources for and the authors make a reasonable empirical case for the utility of the resource.